# Design of a Four-Wheel Steering Mobile Robot Platform and Adaptive Steering Control for Manual Operation

Beomsu Bae [1] and Dong-Hyun Lee [2,*]

1    School of Electronic Engineering, Kumoh National Institute of Technology, 61 Daehak-ro,
     Gumi 39177, Gyeongbuk, Republic of Korea; 20180526@kumoh.ac.kr
2    Department of IT Convergence Engineering, Kumoh National Institute of Technology, 61 Daehak-ro,
     Gumi 39177, Gyeongbuk, Republic of Korea
*    Correspondence: donglee@kumoh.ac.kr; Tel.: +82-54-478-7474

**Abstract:** The recent advancementsin autonomous driving technology have led to an increased utilization of mobile robots across various industries. Notably, four-wheel steering robots have gained significant attention due to their robustness and agile maneuvering capabilities. This paper presents a novel four-wheel steering robot platform for research purposes and an adaptive four-wheel steering control algorithm for efficient manual operation. The proposed robot platform is specifically designed as a simple and compact research-oriented platform for developing navigation and manual operation of four-wheel steering robots. The compact design of the robot platform allows for additional space utilization, while the horizontal independent steering system provides precise control and enhanced maneuverability. The adaptive four-wheel steering control algorithm aims to offer efficient and intuitive manual operation of the four-wheel steering robot, aligning with the intentions of the human operator. It enables the platform to utilize front-wheel steering under normal circumstances and efficiently reduce the turning radius by employing rear wheel steering when additional steering input is required. Experimental results demonstrated the accurate steering performance of the robot platform and effectiveness of the adaptive steering algorithm. The developed four-wheel steering robot platform and the adaptive steering control algorithm serve as valuable tools for further research and development in the fields of autonomous driving and steering algorithms.

**Keywords:** mobile robot platform; four-wheel steering; adaptive steering control; manual operation

## 1. Introduction

With the recent advancements in autonomous driving, there has been a surge in the utilization of mobile robots. Manufacturing and logistics companies predominantly employ mobile robots to mitigate human errors and enhance efficiency and productivity [1–4]. The retail and service sectors, on the other hand, leverage mobile robots for tasks such as baggage handling, customer service, and wayfinding [5,6]. Similarly, in the agricultural domain, mobile robots are employed to address labor shortages and boost productivity [7,8].

To ensure reliable transportation of cargo with varying weights and shapes and stable navigation on uneven road surfaces, four-wheel mobile robot platforms are commonly employed. These platforms encompass skid-steering, two-wheel steering, and four-wheel steering mechanisms. A skid-steering robot adjusts the drive speed of its independently drivable wheels to facilitate movement [9]. However, such robots often encounter wheel slippage during turning due to skid-steering. Omnidirectional robots can be achieved through specialized wheels, such as mecanum wheels [10]. However, these wheels are unsuitable for stable driving due to their complex structure [11]. Two-wheeled steering robots, primarily front-wheel steering, exhibit high-speed capabilities but suffer from spatial inefficiency due to a large turning radius [12].

Four-wheel steering robots offer advantages such as a short turning radius and non-slip rotation of wheels [13,14]. The mechanical structure of four-wheel steering can be

divided into vertical steering and horizontal steering. Vertical steering enables a steering angle exceeding 90°, allowing for independent wheel rotation for turning in place and horizontal movement. However, it presents challenges such as occupying upper space due to the steering motor, limited power, increased control complexity, reduced durability, and difficulties in thermal management of in-wheel motors [15–17]. Horizontal steering, on the other hand, utilizes a single actuator to transmit power to the axles instead of in-wheel motors. While it offers less of a steering angle than vertical steering, it allows for the use of existing chassis without a change in power and offers benefits such as maneuverability, stability, and agility [18,19]. Additionally, in case of a front steering system failure in an autonomous vehicle, the rear wheels can serve as emergency steering, providing an alternative to new redundant designs.

Four-wheel steering platforms can be manually controlled by human operators using a steering unit such as a joystick or steering wheel for precise operation. To accurately control the steering angles during manual operation, factors such as speed changes, weight changes, moments of inertia, and tire friction coefficient can be taken into account [20,21]. While such methods have primarily been applied to typical vehicles with small rear-wheel steering angles, high-speed travel, and mechanical steering, they may not yield significant benefits for mobile robots operating at low speeds for logistics and passenger transportation in urban traffic. In fact, they may lead to decreased steering performance due to increased complexity and inappropriate steering angle determination. Additionally, the implementation of such complex steering algorithms necessitates exhaustive experiments to adjust control parameters appropriately for the target system.

In recent research efforts, several notable contributions have been made towards advancing the capabilities of robotic platforms with four-wheel steering mechanisms. These efforts aim to enhance specific functionalities while addressing certain limitations. In [22], a reconfigurable robot platform is proposed featuring a four-wheel, independently controlled steering and driving mechanism tailored for effective floor cleaning. While the design showcases promise for its intended application, it should be noted that its suitability for broader applications, such as educational and research contexts involving single-body four-wheel steering robots, is limited. This highlights the need for versatile platforms that can accommodate diverse research and educational pursuits. In [23], a comprehensive strategy for seamlessly transitioning between various steering modes, encompassing active front and rear steering, Ackermann steering, and crab steering, is designed to facilitate precise path following. The approach incorporates constrained model predictive control techniques to enable dynamic path tracking for off-road vehicles, accounting for steering and sliding constraints as detailed in [24]. However, they primarily focus on autonomous path following scenarios and lack the adaptability required for intuitive manual control for a human operator. An innovative rear-steering-based decentralized control algorithm (RDC) is presented in [25], aiming to enhance lateral and directional performance for such vehicles. This is further complemented by a robust decoupling controller proposed in [26], which leverages rear-wheel steering alongside cornering stiffness estimation and an inverse model. An additional avenue explored in [27] introduces a control strategy utilizing $\mu$-synthesis, considering tire nonlinearity as parametric uncertainty. It is important to highlight that the primary focus of these works revolves around stability control aspects of four-wheel steering vehicles, particularly addressing challenges such as sideslip suppression and directional stability. However, these investigations have not taken into consideration the development of compact four-wheel steering robot platforms for educational and research purposes, or the intuitive manual control of four-wheel steering by operators.

This paper presents two primary contributions: the introduction of a compact four-wheel steering robot platform, and the development of an adaptive four-wheel steering control algorithm for intuitive manual operation. In light of our investigation, prevailing four-wheel steering robot platforms often suffer from unwieldy size or excessive costs, rendering them unsuitable for deployment within confined indoor spaces for educational and algorithmic testing purposes. While compact front-rear steering platforms are available

in the market, a majority of these systems rely solely on a pair of steering motors, inherently limiting their ability to meet the Instantaneous Center of Rotation (ICR) requirement and lacking integrated differential gears. As a solution, our proposed robot platform features four independently steerable wheels, augmented by an integrated differential gear mechanism. This deliberate design results in a compact and economically viable platform, rendering it suitable for a variety of applications, including education, research, and algorithm validation. The proposed robot platform effectively facilitates empirical inquiries within carefully controlled indoor environments.

The second notable contribution of this study is an adaptive four-wheel steering control algorithm designed to facilitate efficient and intuitive manual operation. The central aim of this steering control algorithm is to enhance vehicle maneuverability and optimize turning performance by adjusting the steering configuration based on the desired steering angle. In cases involving minor target steering angles, the algorithm relies solely on the front steering mechanism while keeping the rear steering fixed. Conversely, when the desired steering angle surpasses a predefined threshold, the algorithm engages the rear steering mechanism to achieve a reduced turning radius. The proposed steering algorithm empowers operators with intuitive manual control over the four-wheel steering robot, eliminating the need for elaborate sensor arrays, complex dynamic models, or intricate optimization algorithms.

This paper is organized as follows. Section 2 describes the proposed four-wheel steering robot platform with detailed specifications and Section 3 presents the proposed adaptive four-wheel steering algorithm. In Section 4, experiments are conducted to validate the accuracy of the four-wheel steering robot platform and the efficiency of the steering control algorithm. The conclusions and future work are described in Section 5.

## 2. Four-Wheel Steering Robot Platform

### 2.1. Robot System Structure

The four-wheel steering robot system consists of a robot module and a remote control module, as shown in Figure 1. The robot module encompasses four steering servo motors, a driving DC motor, an electric speed controller (ESC), a microcontroller, and a ZigBee module. The remote control module consists of a two-axis joystick, a microcontroller, and a ZigBee module.

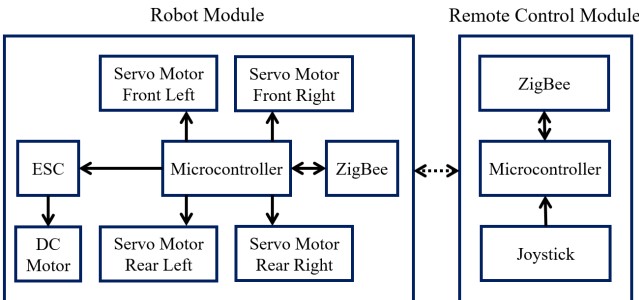

**Figure 1.** Four-wheel steering robot system architecture.

The robot module is equipped with two differential gears, one located at the center of each front and rear axle, connecting the wheels as shown in Figure 2a. These differential gears transfer power from the driving DC motor through the propeller shaft to each wheel, allowing individual wheel rotation at different speeds during robot turns. Through the implementation of four servo motors for horizontal steering, the upper part of the robot frame can accommodate additional sensors and controllers such as LiDAR, a camera, and an embedded controller. The ZigBee in the robot module is utilized for receiving control commands from the remote control module and transmitting the robot's status information, including battery level, current steering angles, and robot speed.

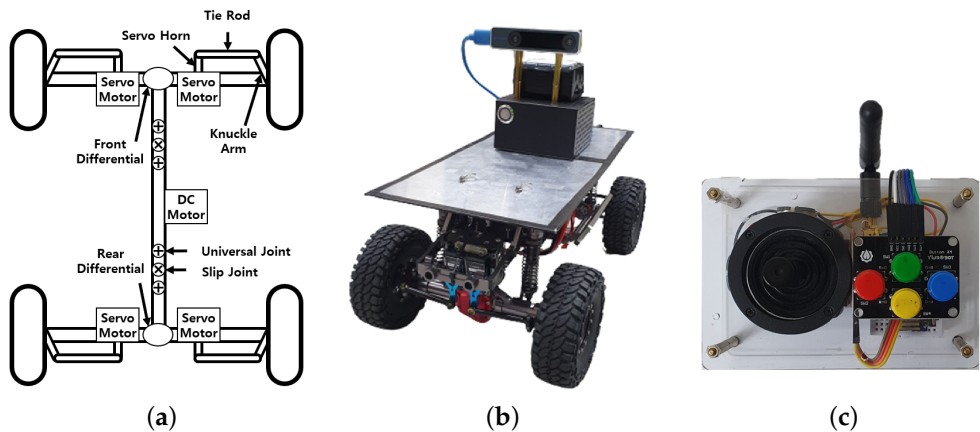

**Figure 2.** (**a**) The structure of robot module, (**b**) overview of robot module, and (**c**) remote control module.

The remote control module consists of a two-axis joystick and four buttons, as shown in Figure 2c. The vertical axis of the joystick controls the linear velocity of the robot, while the horizontal axis determines the steering angle. The four buttons can be mapped to specific actions such as switching operation modes.

### 2.2. Hardware Design of Robot Platforms

The design of the robot platform is specifically tailored for research purposes, focusing on the control and navigation aspects of four-wheel steering robots. Therefore, as indicated in Table 1, it has been intentionally engineered to have a compact size, enabling it to operate effectively within confined indoor or experimental environments. This compact design facilitates precise experimentation and evaluation of control algorithms, navigation strategies, and other related research areas in a controlled setting.

**Table 1.** Detailed specifications of robot platform.

| Parameter | Value |
|---|---|
| Size (L × W × H) | 440 × 240 × 180 mm |
| Weight | 3.8 kg |
| Wheel radius | 110 mm |
| Steering range | $-30°\sim30°$ |
| Maximum speed | 1.5 m/s |
| Minimum turning radius | 501 mm |
| Communication interface | ZigBee |

Table 2 presents the detailed specifications of the electronic components employed in the robot platform. Both the robot module and the remote control module utilize Arduino Nano 33 IoT as the microcontroller. The primary function of the microcontroller in the robot module is the control of steering and speed. Within the remote control module, the microcontroller performs analog-to-digital conversion of the 2-axis joystick's analog values utilizing a 10-bit analog-to-digital converter. It subsequently generates control commands for manual operation. Communication between the microcontrollers in the robot module and the remote control module is facilitated via ZigBee modules. The ZigBee module possesses a bandwidth of 2.4 GHz and operates at a data rate of 250 Kbps. Power to the microcontroller, ESC, and the four servo motors is supplied by the LiPo battery.

### 2.3. Steering Angle Calibration

The steering structure of the proposed robot platform uses servo horns and tie rods for each wheel, as shown in Figure 3a. The process of measuring steering angle as the output of the pulse width modulation (PWM) command of the servo motor is shown in Figure 3b.

The PWM commands of the servo motor and corresponding steering angles of each wheel are collected to approximate a mapping function that maps between the PWM command of the servo motor to the steering angle. Due to the mechanical constraints, the range of the steering angle is set to $-30°\sim30°$.

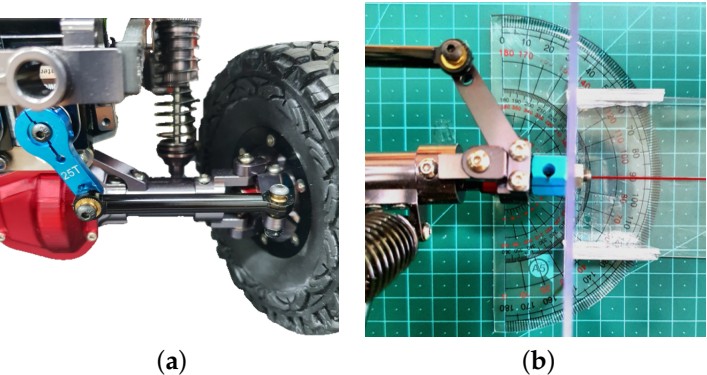

(**a**)                    (**b**)

**Figure 3.** (**a**) Independent wheel steering structure with servo horn and tie rod. (**b**) Measurement of steering angle.

Figure 4 shows the first-order approximations of the four mapping functions. Despite employing identical servo motor models, servo horns, and tie rods, variations in the initial angle of the drive shaft, discrepancies in the servo motor controller, and mechanical imperfections in the power transmission links led to different mapping functions for the four steering servo motors. This indicates that accurate calibration is required for each steering module.

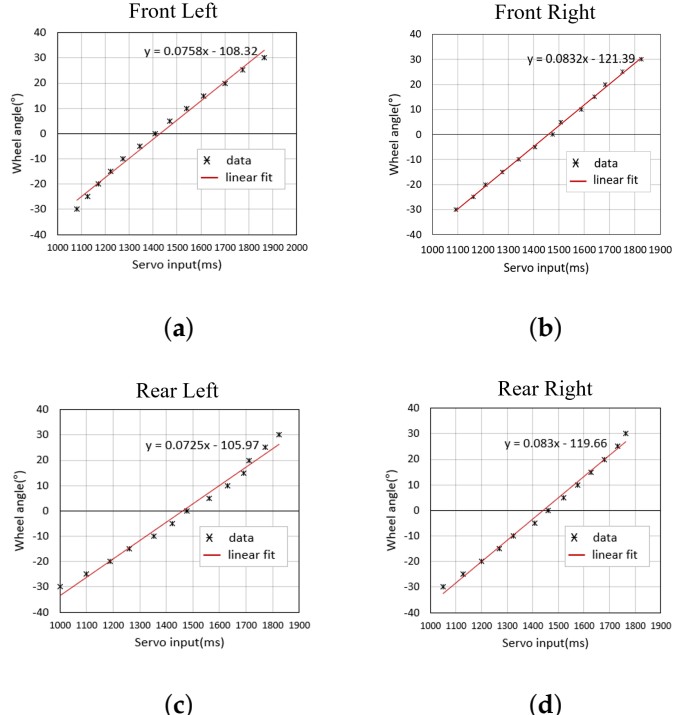

**Figure 4.** First-order approximations of the four mapping functions for (**a**) front left, (**b**) front right, (**c**) rear left, and (**d**) rear right steering servo motors.

**Table 2.** Detailed specifications of electronic components.

| Component Name | Specification |
| --- | --- |
| Arduino Nano 33 IoT | 32 bit microcontroller |
| ESC | Input 2–3 LiPo, continuous current 80 A |
| DC motor | 18,300 rpm, 0.034 Nm |
| Servo motor | 12.0 kg $\times$ cm |
| ZigBee module | ISM 2.4 GHz, 250 Kbps |
| LiPo battery | 11.1 V, 6000 mAh |

## 3. Adaptive Control of Four-Wheel Steering

The adaptive steering control algorithm for manual operation utilizes a combination of front and rear steering to achieve agile maneuverability. By analyzing the target steering angle, the algorithm determines the appropriate steering configuration. The range of the steering angle command $\theta_{cmd}$ is defined as

$$-2\theta_{max} \leq \theta_{cmd} \leq 2\theta_{max} \tag{1}$$

where $\theta_{max}$ is the maximum steering angle that one wheel can be steered.

Figure 5 illustrates the overall process of the proposed adaptive steering control algorithm, where $W$ and $L$ represent the width between the left and right wheels and the wheelbase of the robot, respectively. When the target steering angle is smaller than $\theta_{max}$, indicating a straight or slight turn, the algorithm activates only the front steering, while keeping the rear wheels fixed, as depicted in Figure 5a. In this case, the path of the instantaneous center of rotation (ICR), denoted as $(x^{ICR}, y^{ICR})$, aligns with the extension of the real wheel axle. This configuration provides stability and simplicity during low steering angles.

As the target steering angle exceeds $\theta_{max}$, the algorithm dynamically engages the rear steering, as shown in Figure 5b. In this case, the ICR moves up to the $x$-axis of the robot such that $x^{ICR} = 0$. This effectively reduces the turning radius and improves maneuverability.

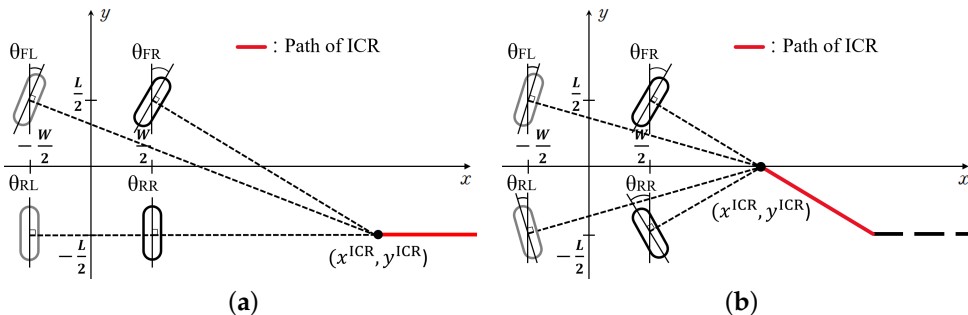

**Figure 5.** Paths of ICR for (**a**) front wheel steering and (**b**) both front and rear wheel steering.

For the right turn, the front-right steering angle $\theta_{FR}$ and rear-right steering angle $\theta_{RR}$ are determined as

$$\theta_{FR} = \begin{cases} \theta_{cmd}, & -\theta_{max} \leq \theta_{cmd} < 0 \\ -\theta_{max}, & -2\theta_{max} \leq \theta_{cmd} < -\theta_{max} \end{cases}$$

$$\theta_{RR} = \begin{cases} 0, & -\theta_{max} \leq \theta_{cmd} < 0 \\ -\theta_{max} - \theta_{cmd}, & -2\theta_{max} \leq \theta_{cmd} < -\theta_{max}. \end{cases} \tag{2}$$

From (2), the location of the ICR is calculated as

$$x^{\text{ICR}} = \frac{W}{2} - \frac{L}{\tan(\theta_{\text{FR}}) - \tan(\theta_{\text{RR}})}, \quad y^{\text{ICR}} = \frac{L}{2} + \tan(\theta_{\text{FR}})(x^{\text{ICR}} - \frac{W}{2}). \tag{3}$$

Using ICR, the front-left steering angle $\theta_{\text{FL}}$ and the rear-left steering angle $\theta_{\text{RL}}$ are determined as

$$\theta_{\text{FL}} = \tan^{-1}(\frac{\frac{L}{2} - y^{\text{ICR}}}{-\frac{W}{2} - x^{\text{ICR}}}), \quad \theta_{\text{RL}} = \tan^{-1}(\frac{-\frac{L}{2} - y^{\text{ICR}}}{-\frac{W}{2} - x^{\text{ICR}}}). \tag{4}$$

In the case of left turn, $\theta_{\text{FL}}$ and $\theta_{\text{RL}}$ are defined as

$$\theta_{\text{FL}} = \begin{cases} \theta_{\text{cmd}}, & 0 < \theta_{\text{cmd}} \leq \theta_{\text{max}} \\ \theta_{\text{max}}, & \theta_{\text{max}} < \theta_{\text{cmd}} \leq 2\theta_{\text{max}} \end{cases}$$

$$\theta_{\text{RL}} = \begin{cases} 0, & 0 < \theta_{\text{cmd}} \leq \theta_{\text{max}} \\ \theta_{\text{max}} - \theta_{\text{cmd}}, & \theta_{\text{max}} < \theta_{\text{cmd}} \leq 2\theta_{\text{max}}. \end{cases} \tag{5}$$

From (5), the location of the ICR is calculated as

$$x^{\text{ICR}} = -\frac{W}{2} - \frac{L}{\tan(\theta_{\text{FL}}) - \tan(\theta_{\text{RL}})}, \quad y^{\text{ICR}} = \frac{L}{2} + \tan(\theta_{\text{FL}})(x^{\text{ICR}} + \frac{W}{2}). \tag{6}$$

Finally, the front-right steering angle $\theta_{\text{FR}}$ and rear-right steering angle $\theta_{\text{RR}}$ are calculated as

$$\theta_{\text{FR}} = \tan^{-1}(\frac{\frac{L}{2} - y^{\text{ICR}}}{\frac{W}{2} - x^{\text{ICR}}}), \quad \theta_{\text{RR}} = \tan^{-1}(\frac{\frac{L}{2} - y^{\text{ICR}}}{\frac{W}{2} - x^{\text{ICR}}}). \tag{7}$$

## 4. Experiments

### 4.1. Steering Accuracy Evaluation

To validate the steering accuracy of the proposed robot platform, a series of experiments were conducted. The experiments involved setting predefined reference circles and measuring the paths followed by the robot. The positions of the five reference circles' ICRs are shown in Figure 6 and Table 3 shows the radius, direction of rotation, and steering angles of four wheels for each of the reference circles. The robot followed each of the given reference circles at a constant velocity of 1 m/s and returned to its starting position.

**Table 3.** Radius, direction of rotation, and steering angles of four wheels for each of the reference circles.

|  | RC$_1$ | RC$_2$ | RC$_3$ | RC$_4$ | RC$_5$ |
|---|---|---|---|---|---|
| Radius (m) | 0.510 | 0.963 | 0.963 | 0.734 | 0.734 |
| Rotation | CCW | CCW | CCW | CW | CW |
| $\theta_{\text{FL}}$ (°) | 20.0 | 0.0 | 20.0 | −5.6 | −16.4 |
| $\theta_{\text{FR}}$ (°) | 15.1 | 0.0 | 17.1 | −6.9 | −20.0 |
| $\theta_{\text{RL}}$ (°) | −20.0 | −20.0 | 0.0 | 16.4 | 5.6 |
| $\theta_{\text{RR}}$ (°) | −15.1 | −17.1 | 0.0 | 20.0 | 6.9 |

In order to capture the robot's path using visual inertial odometry (VIO), an NVIDIA Jetson Nano board and an Intel RealSense T265 were placed at the center of the top plate of the robot platform, as depicted in Figure 2b. T265 includes two fisheye sensors, an inertial measurement unit (IMU), and an vision processing unit (VPU). The visual-simultaneous localization and mapping (V-SLAM) algorithms run directly on the VPU. It provides high precision VIO at a rate of 200 Hz with less than 6 ms latency between movement and

reflection of movement in the pose. It also provides under 1% closed loop drift under intended use conditions.

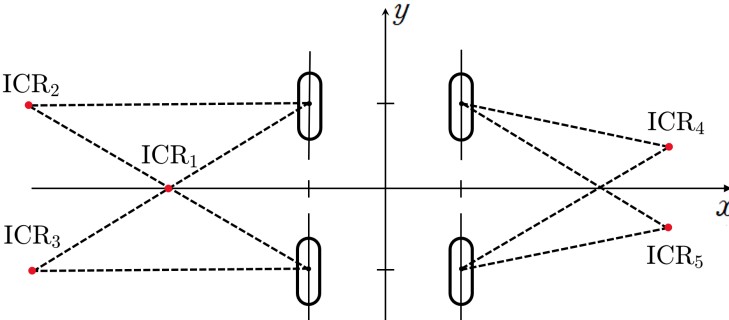

**Figure 6.** Five ICRs of the reference circles.

Figure 7 shows the five reference circles (RC$_1$∼RC$_5$) alongside the corresponding paths of the robot (M$_1$∼M$_5$). The results show that the smaller radius of the reference circle corresponds to a larger error. For a thorough analysis of the results, Figure 8 provides a comprehensive view by illustrating the average and standard deviation of the distance error between the robot's paths and the five reference circles. The position error of the *n*-th position data $(x_n, y_n)$ for the RC$_i$ is defined as

$$e_{i,n} = \left| R_i^{\text{ICR}} - \sqrt{(x_i^{\text{ICR}} - x_n)^2 + (y_i^{\text{ICR}} - y_n)^2} \right| \tag{8}$$

where $R_i^{\text{ICR}}$ is the radius of RC$_i$. Among the paths, M$_1$ exhibits the highest error, with an average of 0.023 m and a standard deviation of 0.012 m, while M$_3$ displays the lowest error, with an average of 0.014 m and a standard deviation of 0.008 m. This relationship arises from several factors, including steering error, VIO error, the condition of the experimental road surface, and the condition of the tires. The error tends to increase as the radius of the reference circle decreases, primarily due to the factors mentioned above.

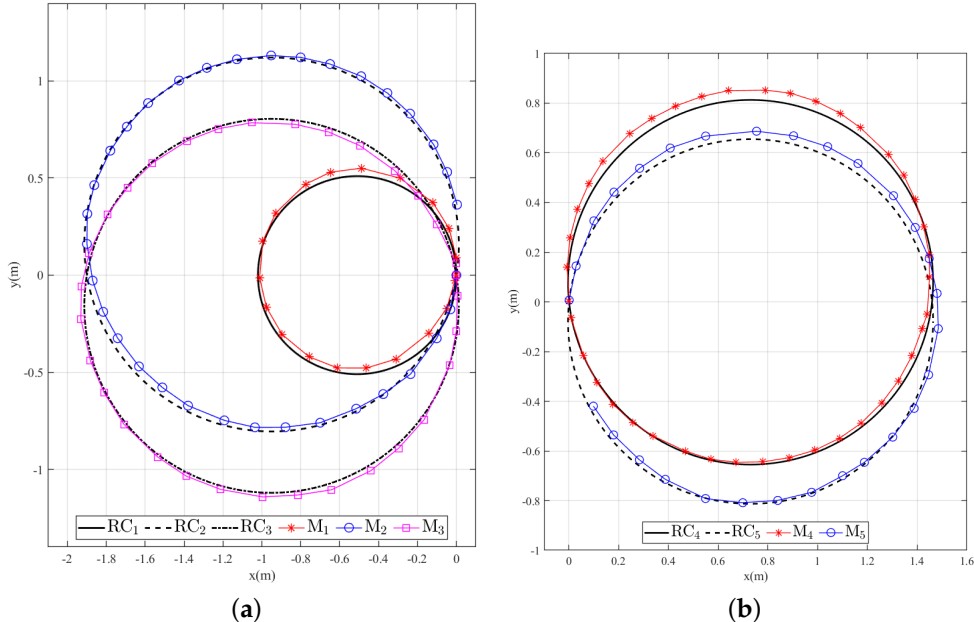

**Figure 7.** Reference circles and corresponding robot paths. (**a**) RC$_1$, RC$_2$, and RC$_3$ with M$_1$, M$_2$, and M$_3$, (**b**) RC$_4$ and RC$_5$ with M$_4$ and M$_5$.

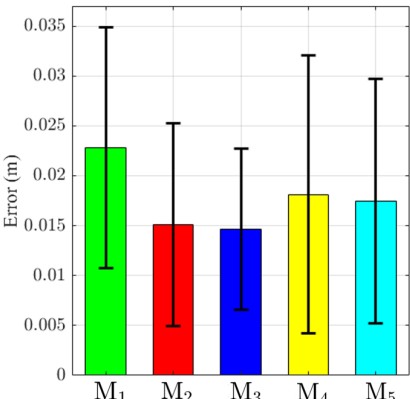

**Figure 8.** Average and standard deviation of the position error by steering algorithm.

### 4.2. Manual Operation Evaluation

To validate the steering efficiency of the proposed adaptive steering control algorithm for manual operation, experiments were conducted where participants utilized a joystick to remotely control the robot. The comparison involved the proposed algorithm and two other steering algorithms: front-wheel steering (FWS) and symmetrical front and the rear wheels steering (SFRWS) algorithms. The FWS algorithm exclusively employs front-wheel steering, while the SFRWS algorithm controls the angle of the rear wheels in counter-phase with the same magnitude angle as the front wheels.

In this experiment, all three steering algorithms were implemented on our robot platform. Specifically, the FWS algorithm was implemented using only front-wheel steering on our robot platform, while the SFRWS algorithm was implemented and applied on the same robot platform, allowing both front and rear wheels to steer in opposite directions.

Figure 9 illustrates the experimental environment, showcasing the goal circle area and obstacles. The study included 10 participants with no previous experience in remotely operating a robot. The objective was to manually operate the robot, avoiding obstacles and reaching the goal circle area in the shortest amount of time. Each participant operated the robot three times using the FWS, SFRWS, and proposed algorithms in a randomized order.

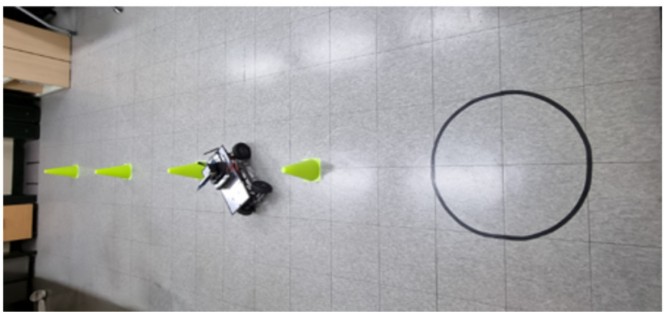

**Figure 9.** Experiment environment with goal area and obstacles.

Figure 10 illustrates the paths of the robot controlled by 10 participants using the FWS, SFRWS, and the proposed steering algorithms. The FWS algorithm exhibited lower steering sensitivity compared to the other algorithms, enabling participants to easily anticipate the robot's direction of motion within the 0 m~1 m segment of the *x*-axis, as depicted in Figure 10a. However, due to its large turning radius, the FWS algorithm necessitated the participants to frequently drive the robot backward to navigate through obstacles, as observed in the 1.5 m~3.5 m segment of the *x*-axis.

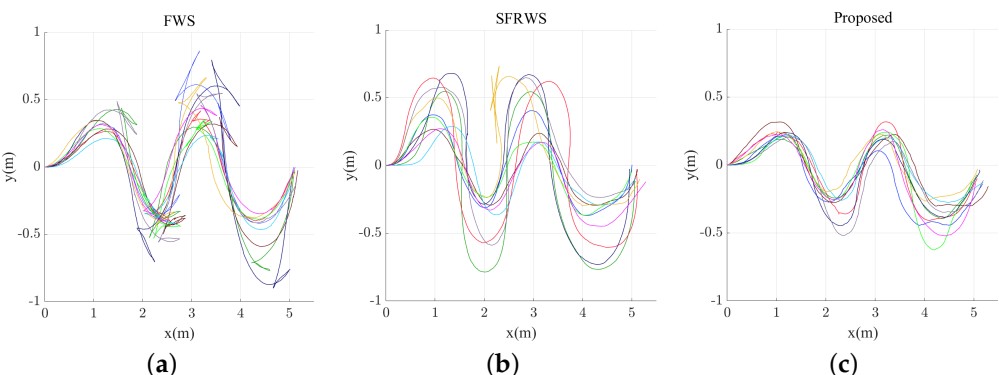

**Figure 10.** Paths of the robot driven by 10 participants using (**a**) FWS, (**b**) SFRWS, and (**c**) proposed algorithms.

In the case of the SFRWS algorithm, the minimum turning radius was half that of the FWS algorithm, allowing for maneuvering in narrow spaces. The robot paths presented in Figure 10b indicate that participants successfully guided the robot through obstacles with fewer reverse maneuvers. However, the large steering sensitivity of the SFRWS algorithm led participants to inadvertently steer the robot farther away from obstacles, resulting in increased arrival time and reduced efficiency. These results suggest that the SFRWS algorithm lacked the necessary intuitiveness to align with participants' manual operation intentions.

The outcomes of the proposed steering algorithm, as depicted in Figure 10c, exhibited reduced deviations and eliminated the need for backward maneuvers. The robot paths during the initial obstacle traversal (0 m~1 m segment of the *x*-axis) resembled those produced by the FWS algorithm. This is due to the proposed algorithm's exclusion of rear wheel steering when only front wheel steering suffices. However, unlike the FWS algorithm, the proposed algorithm adaptively employed rear wheel steering when necessary to maneuver through obstacles without requiring backward motion. The low deviation among the robot paths substantiates the effective conveyance of the participants' intentions through the proposed algorithm.

Figure 11 presents the average and standard deviation of arrival time for the three steering algorithms. The FWS algorithm yielded the largest average and variation of arrival time among participants. Although the SFRWS algorithm exhibited lower average arrival time and reduced variation compared to the FWS algorithm, the deviation in robot paths remained similar, as depicted in Figure 10b. Conversely, the proposed algorithm resulted in the lowest average standard deviation in the arrival time.

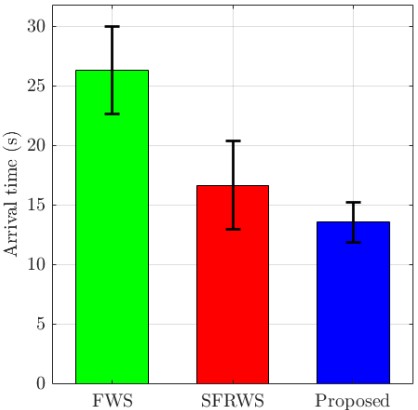

**Figure 11.** Average and standard deviation of arrival time from FWS, SFRWS, and the proposed algorithms.



## 5. Conclusions

In this research, we introduced a four-wheel steering robot platform designed specifically for research purposes, along with an adaptive four-wheel steering control algorithm to facilitate efficient manual operation. The proposed robot platform serves as a valuable tool for the development and validation of navigation and steering control algorithms under diverse scenarios. The adaptive control algorithm, which incorporates front and rear wheel steering, effectively translates the intentions of human operators into manual control of the four-wheel steering robot.

For future works, our objectives include equipping the robot platform with advanced sensors such as LiDAR and stereo cameras to facilitate various applications. Additionally, we aim to enhance the precision and stability of the steering algorithms by incorporating factors such as the load's weight and steering input speed as determinants of the steering angle.

**Author Contributions:** Methodology, B.B. and D.-H.L.; Software, B.B.; Formal analysis, B.B. and D.-H.L.; Investigation, D.-H.L.; Resources, D.-H.L.; Writing—review & editing, B.B. and D.-H.L.; Visualization, B.B.; Supervision, D.-H.L. All authors have read and agreed to the published version of the manuscript.

**Funding:** This research was supported by the Basic Science Research Program of the National Research Foundation of Korea (NRF), funded by the Ministry of Education (2021R1I1A3050100).

**Data Availability Statement:** Data are contained within the article.

**Conflicts of Interest:** The authors declare no conflict of interest.

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
