# Peer review of "Design of a Four-Wheel Steering Mobile Robot Platform and Adaptive Steering Control for Manual Operation"

_electronics, doi:10.3390/electronics12163511_

Round 1
Reviewer 1 Report
This manuscript presents the design of a four-wheel steering mobile robot platform.
However, there have been a huge number of four-wheel steering mobile robots, and some were commercialized. There seems to be little academic contribution of the proposed robot platform. In addition, the used adaptive control method was already proposed a few years ago.
Moreover, the experiment was not well-conducted, since the comparison group was not clearly presented. (Just 2WS vs 4WS vs Proposed. The specific platform should be mentioned for "2WS, 4WS".)
I recommend the author resubmit the manuscript after the establishment of the scientific contribution. Just building the research robot platform has little scientific value for this journal.
Overall English is good.
Reviewer 2 Report
The use autonomous driving technology is proposed of a Four-Wheel Steering Mobile Robot Platform and Adaptive Steering Control for Manual Operation in this work. The manuscript is organized well and contains solid contributions. To further improve the quality of this paper, my detailed comments are given as follows:
1. In the introduction part, the main contribution explanation is missing. More or less work is available in the literature; the authors should compare and explain the main contributions with clear motivation and application of autonomous driving technology.
2. If there is a disturbance or error in the system, will it affect testing and validating the Steering Control for Manual Operation as in many cases, we use sensors, which makes the system much more complicated. What are the difference and difficulties if we use interval observers instead of sensors? Please compare sensors estimation with interval observer estimation; Journal of the Franklin Institute (Volume 358, Issue 6, pp 3077-3126); IEEE Sensors Journal (Volume: 20, Issue: 11, pp 6121 - 6129).
3. There are also language and syntax error issues.
4. More recent resources and comparisons should be cited to complete the bibliography
Overall quality of English is okay, only need proofread.
Reviewer 3 Report
This paper presents a small scale mobile robot capable of four wheel steering and the model necessary to determine required steering angles for individual wheels. The main contribution is a steering strategy which is intuitive for manual control, utilising two-wheel steering for small to moderate steering angles, and four-wheel steering for more extreme steering angles (over 30 degrees). The proposed method is easier to control manually when compared to two wheel steering, or four wheel steering alone.
The main weaknesses are a lack of description of the hardware, and also details on the experimental method. For example, there is no information on details of the steering system geometry, and theoretical capabilities of the system. There are no details on the capabilities of the system in terms of torques and speeds which would be useful to researchers so the system can be understood and incorporated into dynamic vehicle models for autonomous control.
Specific comments:
The introduction/review should be expanded to discuss the control strategies for other similar four-wheel steering mobile robots and highlight the difference in the proposed system.
The introduction states differential drive robots often encounter wheel slippage due to skid steering. It should be made clear here that this is only for skid-steer vehicles with 4 fixed wheels. Differential drive is typically attributed to 2WD robots with caster wheels which are capable of sharp turns and are commonly used for research purposes.
Details of the drive system are lacking. It is difficult to understand the layout of the drive system by the photo in Figure 2 alone.
Calibration of the steering angle is performed experimentally for each wheel - this should be compared to theoretical angles determined from servo angle and geometry of the linkages. How much of the error is attributed to the servo accuracy when compared to play in joints or other sources of error?
The experiment should be described in further detail. What velocity were the tests run at? What are the radii of the desired trajectories, and what steering angles are necessary to achieve the trajectory? Please include a description on how the VIO was collected. The hardware is stated but this provides no information on how the information was processed. State whether the vehicle is moving clockwise or anticlockwise. Please be explicit on how the error of the vehicle position is calculated, presumably it is the euclidean difference between the desired and actual positions. Please quantify how much error is from VIO.
Limitations of the system should be discussed, for example minimum turning radii of the vehicle, lack of feedback for control systems, etc.
R-squared would be useful to show the accuracy of the fits in Figure 4. The data does not appear to be very linear, which could be a result of the steering geometry which is not explored
Please check the equations, particularly Eq. 3 (left). The minus implies that for a right hand turn Px is to the left of the wheel in question.
It is stated that the error in the steering system is negligible relative to the vehicle - please back this up. With an error of 0.023 and SD of 0.012 we can expect 95% of error to be between zero and .047 m. Relative to the vehicle this is ~20% of the width.
It should be made more explicit in the discussion/conclusions that two-wheel steering is more intuitive for manual control, and the adaptive approach provides the best compromise between controlability and turning. The implications for autonomous control should be discussed (for an autonomous system there would not be any difference as the controller would not suffer the same limitation of manual control)
Figure 6 - Some analysis and discussion of the error over time would be useful - it should be expected to increase near the end of the trajectory due to drift.
The information in Fig 8 and 11 could be better presented by removing the bars and using a point to represent the average
Figure 10 would benefit from a plot of steering angle vs time so the range of steering angles during the experiment can be understood
Round 2
Reviewer 1 Report
The authors did a good revision and responses to the first submission, and tried their best to clarify the contributions and experiments.
- The contributions of this paper are 1) to build a compact four-wheel mobile robot and 2) to develop an adaptive four-wheel steering control algorithm to facilitate efficient and intuitive manual operation.
- The robot platforms used for experiments were well-described enough reader to understand the proposed paper's contribution.
I suggest a final check to correct minor typos.